# Descriptive analysis of surveillance data for Zika virus disease and Zika virus-associated neurological complications in Colombia, 2015–2017

**Kelly Charniga**[1]*, **Zulma M. Cucunubá**[1], **Diana M. Walteros**[2], **Marcela Mercado**[2], **Franklyn Prieto**[2], **Martha Ospina**[2], **Pierre Nouvellet**[3], **Christl A. Donnelly**[1,4]

1 Medical Research Council Centre for Global Infectious Disease Analysis, Department of Infectious Disease Epidemiology, Imperial College London, London, United Kingdom, 2 Instituto Nacional de Salud, Bogotá, Colombia, 3 School of Life Sciences, University of Sussex, Brighton, United Kingdom, 4 Department of Statistics, University of Oxford, Oxford, United Kingdom

* kelly.charniga@gmail.com

**Data Availability Statement:** The neurological complications dataset cannot be shared publicly as the data contain information on a small number of

## Abstract

Zika virus (ZIKV) is a mosquito-borne pathogen that recently caused a major epidemic in the Americas. Although the majority of ZIKV infections are asymptomatic, the virus has been associated with birth defects in fetuses and newborns of infected mothers as well as neurological complications in adults. We performed a descriptive analysis on approximately 106,000 suspected and laboratory-confirmed cases of Zika virus disease (ZVD) that were reported during the 2015–2017 epidemic in Colombia. We also analyzed a dataset containing patients with neurological complications and recent febrile illness compatible with ZVD. Females had higher cumulative incidence of ZVD than males. Compared to the general population, cases were more likely to be reported in young adults (20 to 39 years of age). We estimated the cumulative incidence of ZVD in pregnant females at 3,120 reported cases per 100,000 population (95% CI: 3,077–3,164), which was considerably higher than the incidence in both males and non-pregnant females. ZVD cases were reported in all 32 departments. Four-hundred and eighteen patients suffered from ZIKV-associated neurological complications, of which 85% were diagnosed with Guillain-Barré syndrome. The median age of ZIKV cases with neurological complications was 12 years older than that of ZVD cases. ZIKV-associated neurological complications increased with age, and the highest incidence was reported among individuals aged 75 and older. Even though neurological complications and deaths due to ZIKV were rare in this epidemic, better risk communication is needed for people living in or traveling to ZIKV-affected areas.

## Introduction

Zika virus disease (ZVD) is an emerging infectious disease caused by Zika virus (ZIKV), a single-stranded RNA virus that belongs to the genus *Flavivirus* [1]. In 1952, the first cases of ZVD in humans were reported in present-day Tanzania. The bite of infected *Aedes* spp. mosquitoes,

patients. Although the data are anonymized, identification is a risk given the high geographic resolution and large combination of predictor variables. This determination was made by Comité de Ética y Metodologías de Investigación (CEMIN). To request access to these data, please contact: secretariactin-cein@ins.gov.co, (57+1) 2207700 Ext. 1331-1108, Colombia. The aggregated ZIKV dataset is available on GitHub: https://github.com/kcharniga/descriptive_zika. The data include the number of weekly reported cases by administrative level 2 (municipality) as well as sex and age category.

**Funding:** We acknowledge funding from the MRC Centre for Global Infectious Disease Analysis (reference MR/R015600/1), jointly funded by the UK Medical Research Council (MRC) and the UK Foreign, Commonwealth & Development Office (FCDO), under the MRC/FCDO Concordat agreement and is also part of the EDCTP2 programme supported by the European Union. KC is funded by Imperial College London's President's PhD Scholarship. ZMC is supported by a Fellowship through the Rutherford Fund (MR/R024855/1). The funders had no role in study design, data collection and analysis, decision to publish, or preparation of the manuscript.

**Competing interests:** The authors have declared that no competing interests exist.

particularly *Ae. aegypti*, is the main route of ZIKV transmission in urban epidemics [2]. Human-to-human transmission can occur through sex and transfusion of infected blood products as well as from mother to child during or after pregnancy [3–5]. Symptoms of ZVD include fever, rash, joint pain, muscle aches, conjunctivitis, and headache. Severe cases and deaths are rarely reported [6], and about 75%-80% of persons infected by ZIKV do not show any symptoms [7]. No approved medical countermeasures exist to prevent or treat ZIKV infection [8], but several vaccine candidates are under evaluation in pre-clinical through phase 2 studies [9].

Historically, ZIKV was thought to cause only mild illness in humans [2]. During the 2010s, major ZIKV epidemics in Micronesia, French Polynesia, the Caribbean, and Latin America demonstrated that a small proportion of ZIKV-infected individuals experience serious disease, including Guillain-Barré syndrome (GBS). GBS is an autoimmune condition with a global annual incidence estimated at 1.1 to 1.8 cases per 100,000 population [10]. It is typically preceded by a viral or bacterial infection, especially *Campylobacter jejuni*, and is the most common cause of non-poliovirus acute flaccid paralysis globally [11]. GBS has also been associated with certain vaccines, including vaccines for rabies, tetanus, and influenza [11]. In 2013–2014, a spike in the number of GBS cases was detected in French Polynesia during the largest documented ZIKV outbreak at that time [12]. Since then, evidence of an association between ZIKV infection and GBS has continued to increase [13, 14].

Symptoms of GBS include tingling, numbness, or pain in the limbs as well as limb weakness. Most patients with GBS require hospitalization and some require intensive care and ventilatory support [15]. Between 3–10% of GBS patients die from the condition [16]. Although most patients fully recover, some may experience long-term morbidity, including depression and disability [17, 18]. Treatment for acute GBS involves the administration of intravenous immunoglobulin and plasma exchange [18].

Research suggests that the risk of GBS tends to be higher for males than females and increases with age [11]. According to a meta-analysis of population-based studies of GBS in North America and Europe, the relative risk for males versus females was estimated at 1.78 (95% CI: 1.36–2.33). The study also found that the rate of GBS increased 20% for each 10-year increase in age [11]. There is evidence of seasonal variation in GBS with most studies indicating higher incidence during winter (January to March) compared to the other three seasons. A meta-analysis found geographical heterogeneity across published studies which could be due to regional differences in the seasonality of infections that trigger GBS [19].

In addition to GBS, ZIKV infection during pregnancy has been associated with congenital ZIKV syndrome (CZS) in fetuses and newborns. CZS is characterized by microcephaly, decreased brain tissue, eye damage, limited range of motion in the joints, and excessive muscle tone that restricts movement [20]. Most newborns with prenatal exposure to ZIKV do not develop clinical signs of CZS [21]. Yet, cohort studies have shown that children without birth defects who were exposed to ZIKV in utero can still experience neurological problems and developmental delays during the first two years of life [22, 23]. For these reasons, pregnant females are considered a high-risk group for ZIKV infection. Infants and children can also become infected with ZIKV during the postnatal period through mosquito bites and possibly breast milk; however, few studies have evaluated postnatal ZIKV infection prospectively [24].

Colombia was one of the countries most affected by the 2015–2017 ZIKV epidemic in the Americas. Surveillance for ZVD began in August 2015. By December 2015, GBS cases and other neuroinflammatory disorders began to rise in the country [14]. From the end of January to mid-November 2016, the number of reported microcephaly cases in Colombia increased fourfold compared to the same time period in 2015 [25]. Here, we describe epidemiological trends of ZVD and ZIKV-associated neurological complications in Colombia. Cumulative

incidence, risk ratios, and tests for statistical significance were calculated for high-risk groups. Understanding risk factors for neurological complications could inform prevention efforts and improve interpretation of ZIKV surveillance data.

## Methods

### Ethics statement

The technical and ethical endorsement of this study was provided by the Comité de Ética y de Metodologías de Investigación (CEMIN) of the Instituto Nacional de Salud (INS) of Colombia (project number 35–2017).

### Data

We analyzed two anonymized line lists on ZVD cases and ZIKV-associated neurological complications. The location of cases used in this study refers to location of likely infection, which is determined by the clinician who reported the case.

**ZIKV.** The ZIKV dataset consists of 106,033 suspected and laboratory-confirmed cases of ZVD reported to Sivigila, Colombia's national public health surveillance system, between 2015 and 2017. We included cases with missing information on municipality (administrative level 2) location. Information related to public health events in the national territory is generated from the local levels by local health service providers. There are roughly 14,000 institutional, municipal, departmental, or national reporting bodies in Colombia. ZVD was added to the list of notifiable conditions in 2015. Each week these data are aggregated and published [26].

The line list data were aggregated by municipality. Data were further aggregated by week based on either date of symptom onset or date of notification. For ZIKV, this resulted in a time series spanning 97-weeks, from the week ending August 15, 2015 to that ending June 17, 2017, or epidemiological week 32 of 2015 to epidemiological week 24 of 2017.

**Neurological complications.** The neurological complications dataset contains information on 418 patients with neurological complications and recent history of febrile illness compatible with ZVD. It includes those with GBS as well as other conditions such as myelitis and meningoencephalitis but excludes cases with microcephaly and other congenital defects. Four cases in this dataset were laboratory-confirmed for ZIKV by reverse-transcriptase PCR.

The number of cases with neurological complications here is a few hundred smaller than previously published data from Colombia [27, 28]. Throughout the epidemic, cases of neurological complications associated with prior ZIKV infection were reported in the INS Weekly Epidemiological Bulletin. This information was made publicly available with the caveat that cumulative case numbers could change following a verification process [28].

Medical records of patients with neurological complications were reviewed using case definitions from the Brighton Collaboration Working Group for GBS, myelitis, encephalitis, and acute disseminated encephalomyelitis (ADEM) [29, 30]. With the goal of improving comparability of vaccine safety data, the Brighton Collaboration developed standard case definitions and guidelines for neurologic adverse events following immunization. The criteria could be applied to a range of settings, including across geographical regions as well as different levels of healthcare quality and access. Case definitions are organized according to three levels of diagnostic certainty, from Level 1 (most certain) to Level 3 (least certain) [31]. Patients that did not meet Brighton case definition criteria 1–3 were removed from the dataset.

Both date of symptom onset of neurological complications and date of notification were available for all cases in the neurological complications' dataset. Dates corresponding to symptom onset spanned 122 weeks, from the week ending July 4, 2015 to that ending October 28, 2017 (epidemiological week 26 of 2015—epidemiological week 43 of 2017). Notification dates

spanned 108 weeks, from the week ending October 17, 2015 to that ending November 4, 2017 (epidemiological week 41 of 2015—epidemiological week 44 of 2017). Data were aggregated by week.

**Demographic data.**   Population demographic data for 2016 were obtained from the Departamento Administrativo Nacional de Estadística (DANE), Colombia's National Administrative Department of Statistics. These data consist of population projections derived from the 2005 Census.

### Descriptive analysis

The cumulative incidence (attack rate) of ZVD was estimated using DANE population projections for 2016 as the denominator. Unless otherwise noted, the cumulative incidence of ZIKV-associated neurological complications was estimated using reported cases of ZVD as the denominator. For incidence by geographic location, the total population was included for each reporting area regardless of altitude (except for the capital city of Bogotá). Confidence intervals were calculated using the binomial exact function in the R package epitools (version 0.5–10.1). Risk ratios (RR) and 95% confidence intervals were calculated using the riskratio function in the R package fmsb (version 0.7.0).

In order to estimate the incidence of ZVD by pregnancy status, it was necessary to estimate the number of pregnant females and non-pregnant females in Colombia. In 2017, the Colombian Ministry of Health estimated the annual number of pregnant females for 2017–2019. The mean estimate of 822,396 for 2017 was multiplied by ¾ (females are only pregnant for ¾ of the year on average) to obtain the number of pregnant females in the population at any time (616,797). To obtain the number of females who were not pregnant at any time, this number was subtracted from the projected total number of females in the population in 2016 (24,678,673, from DANE).

In the ZIKV dataset, 1,636 females had missing pregnancy status.

## Results

### ZIKV

**Sex and age trends.**   More ZVD cases were reported in females than males each year from 2015–2017 (Table 1). Overall, females represented two-thirds (66.2%) of reported cases, which differs significantly from the general population of Colombia (50.6% female [exact binomial test, p < 0.001]). The RR of ZVD in females was nearly two times higher than in males (1.91, 95% CI: 1.88–1.93).

The median age of ZVD cases in Colombia was 29 years (range 0 to >100 years). Nearly half (49.0%) of cases were reported in individuals between the ages of 20 and 39. The number of cases reported in this age group was significantly different than expected compared to the general population of the same age group (31.2% [exact binomial test, p < 0.001]).

**Table 1.  Number of ZVD cases by sex in Colombia.**  Epidemiological week 32 of 2015—epidemiological week 24 of 2017.

| | Female | | Male | | Total | |
|---|---|---|---|---|---|---|
| | N | % | N | % | N | % |
| *2015* | 8,940 | 63.7 | 5,085 | 36.3 | 14,025 | 13.2% |
| *2016* | 60,494 | 66.7 | 30,153 | 33.3 | 90,647 | 85.5% |
| *2017* | 738 | 54.2 | 623 | 45.8 | 1,361 | 1.3% |
| *Total* | 70,172 | 66.2 | 35,861 | 33.8 | 106,033 | 100% |

The cumulative incidence of ZVD was significantly higher for females compared to males across all age groups except 0–4 years and those 80 years and over (Fig 1). Among those 15 to 29 years of age, the risk of ZVD was about three times higher in females compared to males. The highest RR of female to male cases was observed in the 20 to 24-year age group (3.16, 95% CI: 3.04–3.28).

**Pregnancy.**   A total of 19,243 ZVD cases was reported in pregnant females. Pregnant females had much higher risk of being reported as a ZVD case compared to males and non-pregnant females. The cumulative incidence of ZVD in pregnant females was 3,120 reported cases per 100,000 population (95% CI: 3,077–3,164). In contrast, the incidence in non-pregnant females was 205 cases per 100,000 population (95% CI: 203–207), and the incidence in males was 149 cases per 100,000 population (95% CI: 147–151). The differences in incidence of ZVD by sex and pregnancy status were significantly different.

**Geographical distribution.**   Cases of ZVD were reported in all 32 departments of the country. The departments with the highest number of reported ZVD cases included Valle del Cauca (27,712), followed by Santander (10,374) and Norte de Santander (10,361). The highest incidence of ZVD per 100,000 population was reported in San Andrés and Providencia (1,489), followed by Casanare (1,087) and Norte de Santander (758) (Fig 2).

**Temporal trends.**   The peak of the ZIKV epidemic occurred during the week ending on February 6, 2016. For pregnant females, the number of cases peaked one week earlier than both males and non-pregnant females, in the week ending on January 30, 2016. Fig 3 shows the epidemiological curves of ZVD cases at the national level for Colombia. Chikungunya fever (CF) cases are also shown for reference. The figure shows that the number of reported

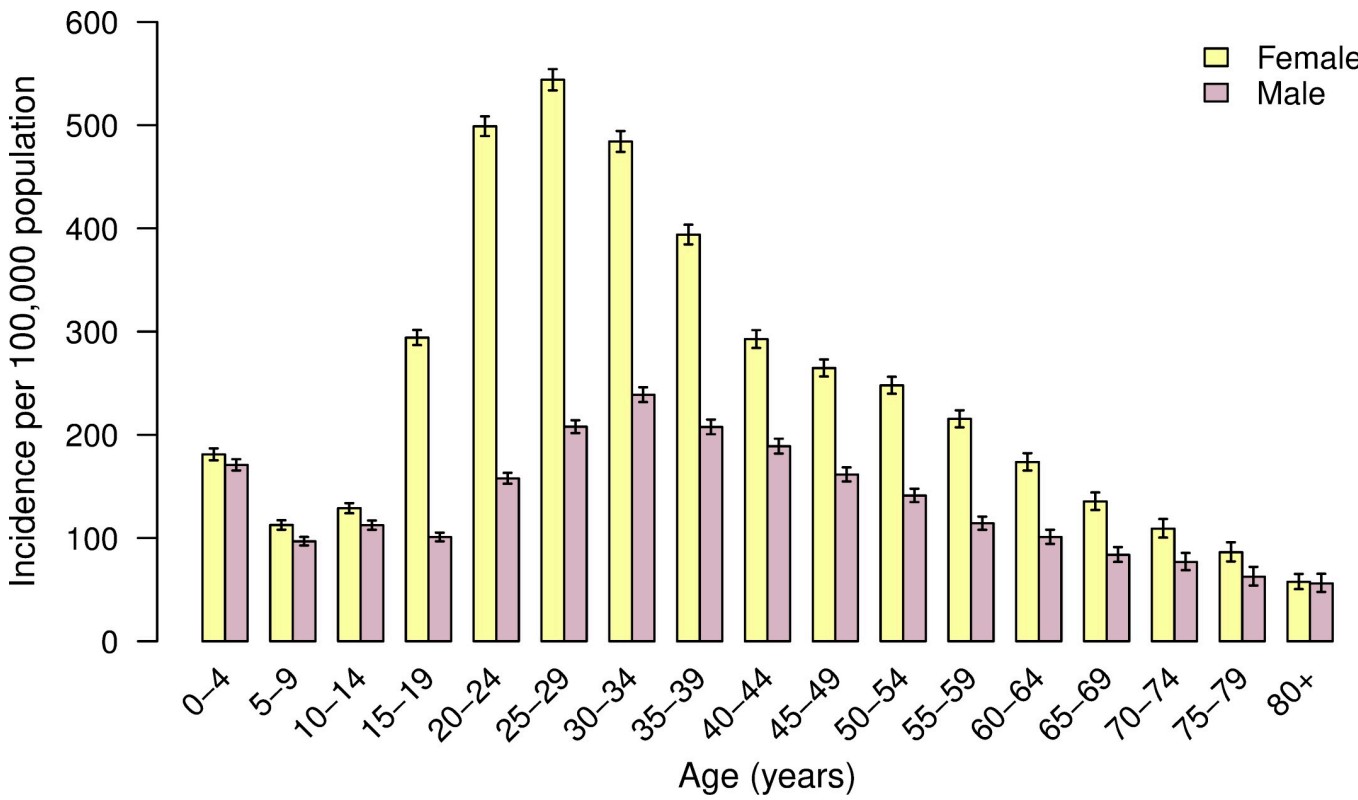

**Fig 1. Cumulative incidence of ZVD per 100,000 population by age group and sex in Colombia.** Epidemiological week 32 of 2015—epidemiological week 24 of 2017.

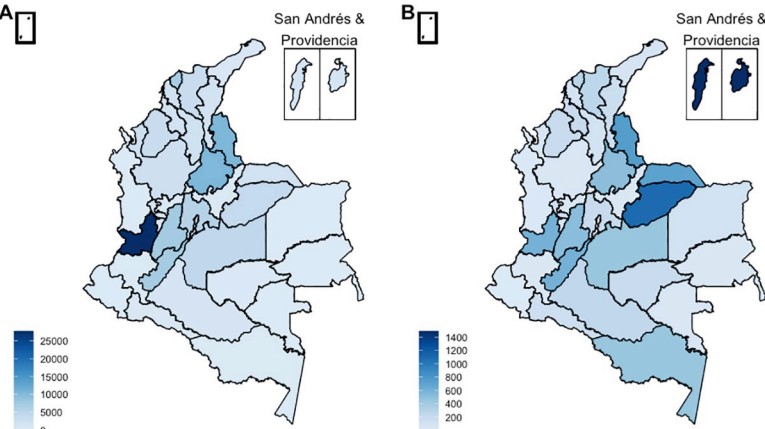

**Fig 2. Maps of ZVD incidence in Colombia.** (A) Number of cases (total N = 106,033). (B) Cumulative incidence of ZVD per 100,000 population. Maps were created with shapefiles from the Humanitarian Data Exchange [32]. The shapefiles were prepared by OCHA and are available under a CC BY-IGO license [33]. No changes were made to the shapefiles.

CF cases decreased in the week prior to both New Year's Day and Easter in 2015 and subsequently increased. In 2016, a similar pattern occurred for reported ZVD cases.

The epidemiological curves of ZVD cases are shown for all departments in Fig 4. Most departments had one epidemic peak between January and April 2016, but some appeared to

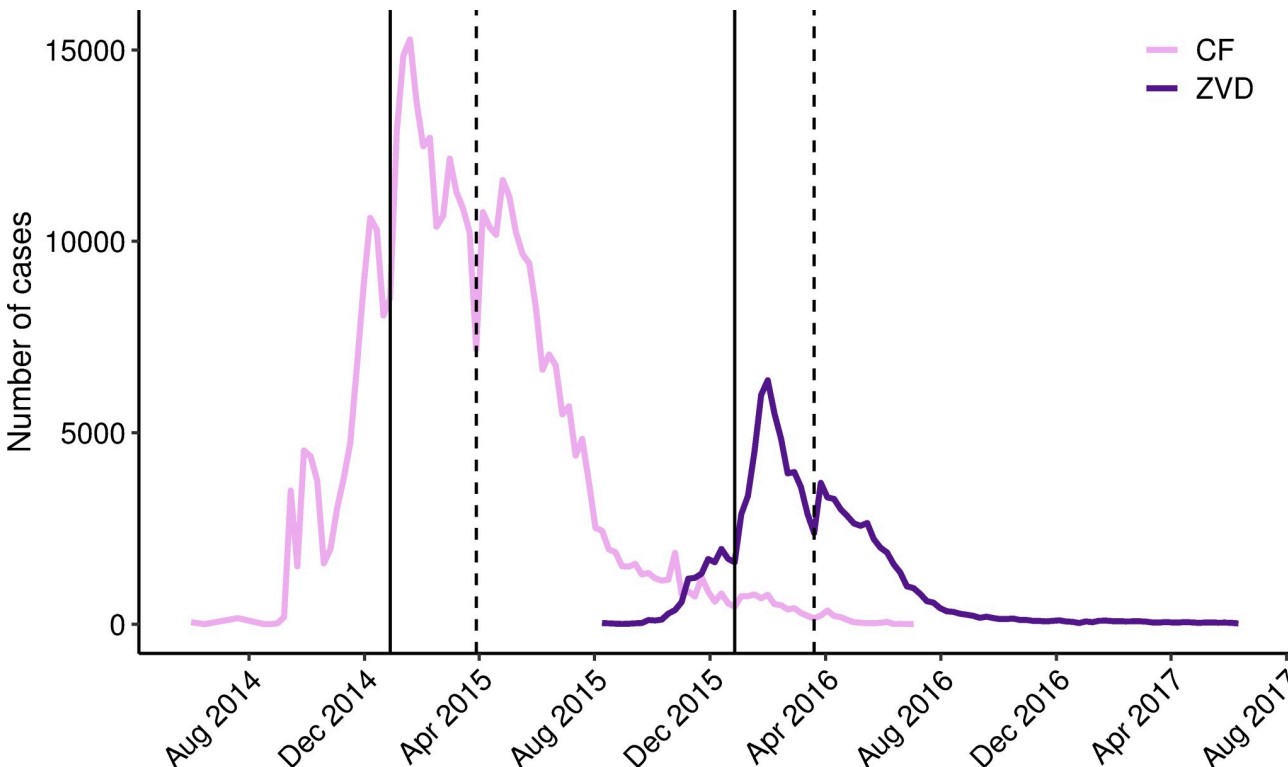

**Fig 3. Number of chikungunya fever (CF) and Zika virus disease (ZVD) cases reported each week during the 2014–2017 epidemic in Colombia.** Dotted lines mark the weeks preceding Easter in 2015 (epidemiological week 13) and 2016 (epidemiological week 12). Solid lines mark the weeks preceding New Year's Day in 2015 (epidemiological week 53 of 2014) and 2016 (epidemiological week 52 of 2015).

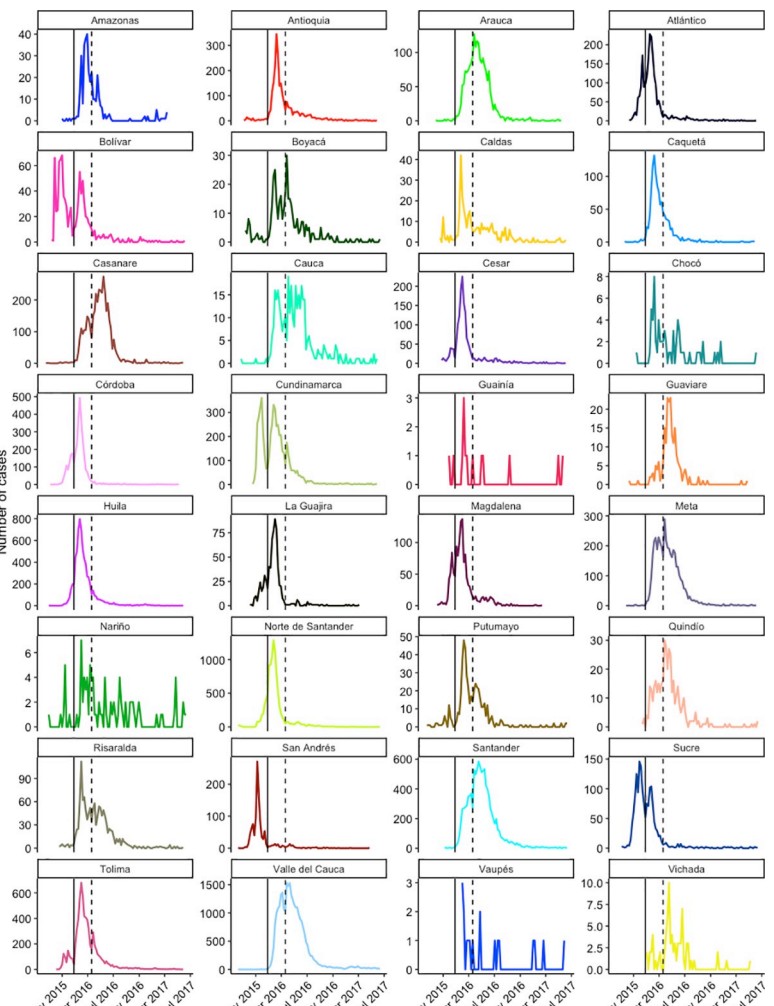

**Fig 4. Number of ZVD cases reported each week by department during the 2015–2017 epidemic in Colombia.**
Dotted lines mark the week preceding Easter 2016 (epidemiological week 12). Solid lines mark the week preceding
New Year's Day in 2016 (epidemiological week 52). Y-axes are different, and x-axes are the same.

have two peaks (Bolívar and Cundinamarca). Chocó, Guainía, Nariño, Vaupés, and Vichada
had irregular time series due to small numbers of reported cases.

## Neurological complications

**Sex and age trends.** The median age of ZVD cases with neurological complications was
41 years (range 0–93). The highest number of neurological complications was reported in mid-
dle-aged groups from 35–49, and the cumulative incidence of neurological complications was
highest in those at least 75 years of age with a high degree of uncertainty.

Two-hundred and forty-four cases (58%) were male. In the general population, the risk of
neurological complications was 44% higher in males than females (RR = 1.44, 95% CI: 1.18–
1.75). Among reported ZVD cases, the risk of neurological complications was 174% higher in
males compared to females (RR = 2.74, 95% CI: 2.26–3.33).

For both sexes, the greatest number of complications was reported in middle-aged groups.
The cumulative incidence of neurological complications per 1,000 cases of ZVD by age and

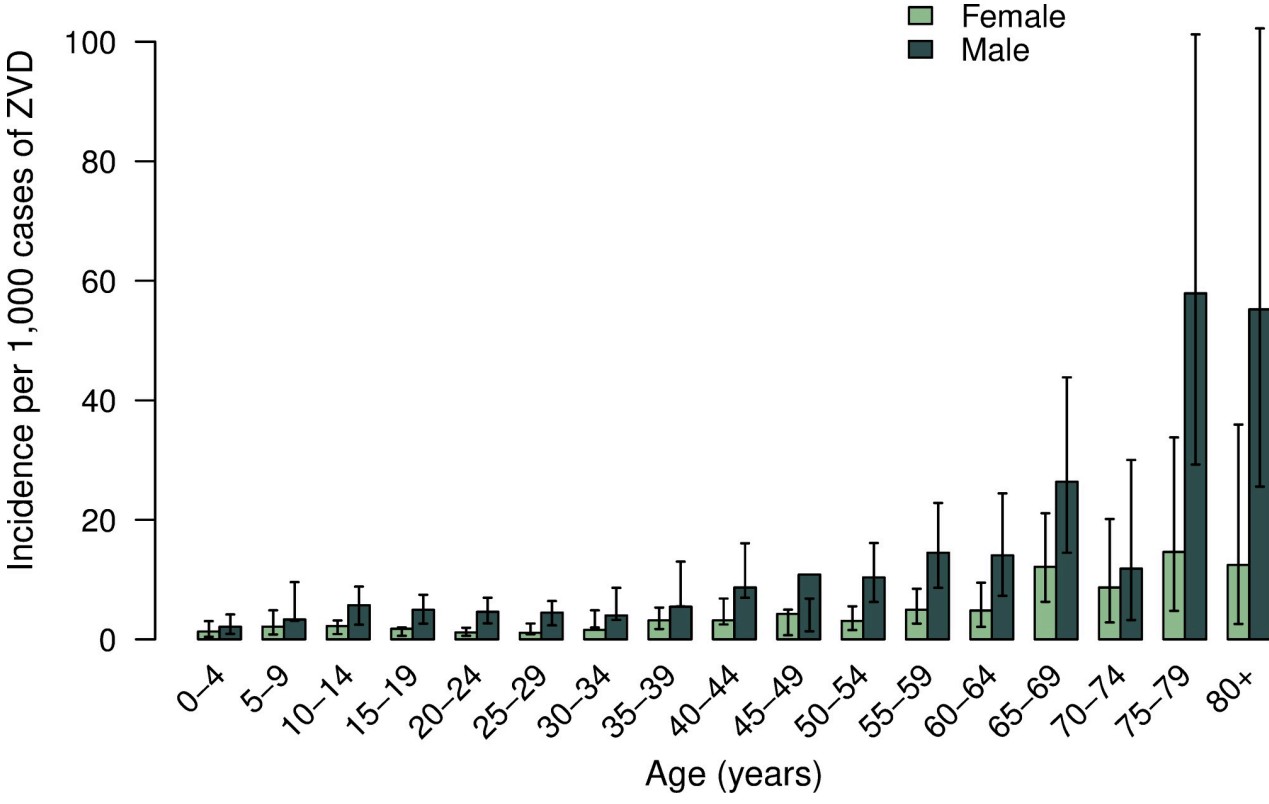

**Fig 5. Cumulative incidence of ZIKV-associated neurological complications per 1,000 cases of ZVD by age group and sex in Colombia.**

sex is shown in Fig 5. For both sexes, the incidence of neurological complications increased with age. Males aged 75 and older had the highest incidence but also the most uncertainty, which is reflected in the wide confidence intervals.

**Diagnosis and final condition.** Three hundred and fifty-two patients (85%) with ZIKV-associated neurological complications were diagnosed with GBS. Encephalitis and myelitis were the second and third most common diagnoses, respectively, with 29 and 17 cases each. Facial paralysis was recorded for 11 patients. Meningitis (N = 2), meningoencephalitis (N = 2), and optic neuritis (N = 1) were the least commonly diagnosed complications.

Thirty-five cases (8.4%) resulted in death.

**Geographical distribution.** Twenty-eight out of 32 departments in Colombia were reported as the location of likely infection for at least one case of ZIKV-associated neurological complications (Fig 6). The highest number of cases was reported in Atlántico with 104. Norte de Santander and Valle del Cauca had the next highest number of cases with 61 and 44, respectively. The highest cumulative incidence of ZIKV-associated neurological complications per 1,000 cases of ZVD was reported in Nariño followed by Chocó and Atlántico (Table 2 and Fig 6). However, 24 departments had 10 cases or fewer, and the estimated incidence should therefore be interpreted with caution.

At the municipality level, Barranquilla was reported as the location of likely infection for the highest number of ZIKV-associated neurological complications with 80, followed by Cúcuta with 44 and Cali with 23. Cases were spread out geographically and tended to cluster in large cities.

**Temporal trends.** The distributions of ZVD cases and cases of ZIKV-associated neurological complications over time are similar. There are differences, however, in the time series of

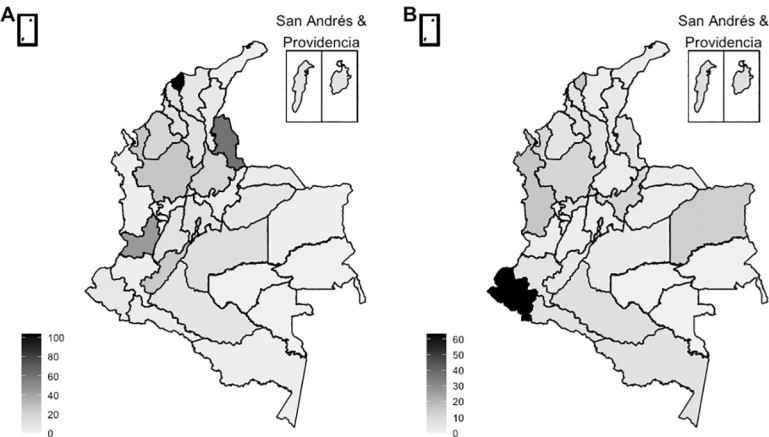

**Fig 6. Maps of ZIKV-associated neurological complications by department in Colombia.** (A) Number of cases. N = 406 observations with non-missing location at department level. (B) Cumulative incidence of ZIKV-associated neurological complications per 1,000 ZVD cases. Maps were created with shapefiles from the Humanitarian Data Exchange [32]. The shapefiles were prepared by OCHA and are available under a CC BY-IGO license [33]. No changes were made to the shapefiles.

**Table 2. Cumulative incidence and 95% confidence intervals of ZIKV-associated neurological complications per 1,000 ZVD cases by department in Colombia.**

| Department | Incidence | 95% CI | |
|---|---|---|---|
| Nariño | 63.2 | 23.5 | 132.4 |
| Chocó | 15.4 | 0.4 | 82.8 |
| Atlántico | 15.3 | 12.6 | 18.6 |
| Vichada | 12.8 | 0.3 | 69.4 |
| Boyacá | 10.9 | 3.0 | 27.7 |
| Antioquia | 10.3 | 6.7 | 15.1 |
| Caquetá | 7.0 | 3.0 | 13.8 |
| San Andrés & Providencia | 7.0 | 3.0 | 13.7 |
| Sucre | 6.1 | 2.9 | 11.2 |
| Caldas | 6.0 | 0.7 | 21.5 |
| Putumayo | 6.0 | 1.2 | 17.4 |
| Norte de Santander | 5.9 | 4.5 | 7.6 |
| Amazonas | 5.8 | 0.7 | 21.0 |
| Cauca | 5.6 | 0.7 | 20.2 |
| Bolívar | 5.2 | 2.5 | 9.6 |
| Córdoba | 5.1 | 3.0 | 8.2 |
| Huila | 3.4 | 2.2 | 5.1 |
| Cesar | 3.1 | 1.0 | 7.3 |
| Meta | 2.8 | 1.4 | 4.8 |
| Quindío | 2.5 | 0.1 | 13.6 |
| Magdalena | 2.2 | 0.9 | 4.5 |
| Arauca | 2.1 | 0.6 | 5.5 |
| Casanare | 1.8 | 0.7 | 3.7 |
| Santander | 1.7 | 1.0 | 2.7 |
| Cundinamarca | 1.7 | 0.8 | 3.2 |
| Valle del Cauca | 1.6 | 1.2 | 2.1 |
| La Guajira | 1.4 | 0.0 | 7.9 |
| Tolima | 1.3 | 0.6 | 2.4 |
| Guainía | 0.0 | 0.0 | 231.6 |
| Vaupés | 0.0 | 0.0 | 185.3 |
| Guaviare | 0.0 | 0.0 | 17.6 |
| Risaralda | 0.0 | 0.0 | 2.8 |

neurological complications when date of symptom onset is considered rather than date of notification. The week with the highest number of neurological complications according to date of symptom onset was the same week in which the most ZVD cases were reported (30 cases during the week ending on February 6, 2016). In contrast, the most neurological complications according to date of notification were reported two weeks earlier, in the week ending on January 23, 2016 (33 cases).

Similar to the ZIKV dataset, which did not have date of symptom onset for all cases, date of notification (reporting) of ZIKV-associated neurological complications seemed to decrease during the Christmas/New Year and Easter holidays.

Although the distribution of neurological complications by sex did not vary across time, more cases were consistently reported in males than in females throughout the epidemic.

## Discussion

This work builds on a preliminary report of ZVD cases in Colombia that also used data from the national population-based surveillance system [26]. The earlier report was published just after the peak of the epidemic and included 65,726 cases reported between August 9, 2015 and April 2, 2016. This analysis adds an additional 40,307 cases reported until June 17, 2017 and data on severe cases with neurological complications, including GBS.

### ZIKV

Compared to the general population of Colombia, ZVD cases were more likely to be reported in individuals in their 20s and 30s. Several factors can affect the age distribution of cases found through epidemic surveillance, including age-related variation in susceptibility, reporting bias, pre-existing immunity, the age distribution of the population, and level of exposure to infection [34].

Seroprevalence studies, which test for antibodies indicative of past infection, can be used to assess age-related variation in susceptibility. Although at least one such study found a positive association between ZIKV infection and age, several studies in different countries have found no significant association [35–40].

Based on the timing and origins of ZIKV arriving in the Americas from Southeast Asia and the Pacific [41], no immunological protection for ZIKV was assumed at the population level prior to this epidemic. If there was pre-existing immunity, lower infection rates would have been expected in older age groups that had been exposed in the past assuming long-lasting immunity.

Risk factors for ZIKV infection are poorly understood [37]. On Yap Island, Duffy et al. found no behavioral risk factors for ZIKV infection [35]. In contrast, Lozier et al. found higher prevalence of ZIKV among those who reported being bitten by mosquitoes at home in bivariate analyses, but the association was not statistically significant in multivariable analysis [36].

In this study, females had higher cumulative incidence of reported ZVD than males. Case ascertainment was likely higher in females of child-bearing age due to concerns about birth defects. However, reporting bias does not explain the elevated risk in females versus males between the ages of 45 and 79. This result could be explained by differences in susceptibility or exposure. If females in this age range spend more time at home than their male counterparts, they might experience higher exposure to *Ae. aegypti* mosquitoes, which tend to live in and around people's homes in urban areas [42, 43]. The higher incidence of ZVD in females is consistent with ZIKV epidemics in other locations such as Brazil, Puerto Rico, and Yap [35, 36, 44]. Spending more time at home might also increase the risk of exposure to mosquito bites

and therefore ZIKV infection in young children, which could explain the higher incidence of ZVD in the youngest age group (0–4 years) compared to older children in this study.

Notification of ZVD cases decreased during Christmas/New Year and Easter holidays in 2015–2016. This trend could be seen at the national and subnational level as well as during the CHIKV epidemic. Changes in the number of patient consultations for general practice services on and immediately after public holidays have been observed in several countries, including the U.K. This has been called the "public holiday effect" [45]. In Colombia, where 93% of the population identifies as Christian, surveillance for notifiable diseases is likely impacted by religious holidays [46].

## Neurological complications

Despite higher cumulative incidence of ZVD cases in females compared to males, higher incidence of neurological complications was observed in males, a finding which is consistent with GBS epidemiology and other studies [47, 48].

The median age of ZVD cases with neurological complications was 12 years older than that of ZVD cases. The cumulative incidence of ZIKV-associated neurological complications increased with age, and the highest incidence was reported among individuals aged 75 and older. The positive association between age and neurological complications here is also consistent with GBS epidemiology [47].

Time of symptom onset of ZIKV infection was not available for all cases who went on to develop neurological complications in this dataset. According to a 2016 study of 71 patients in Puerto Rico, the median time between onset of ZIKV infection symptoms and GBS was seven days (range 0–21 days) [13]. Similarly, a 2015–2016 study reported that the median time between onset of ZIKV infection symptoms and GBS symptoms was 7 days (interquartile range, 3–10 days) for 66 patients from six Colombian hospitals [14].

The most common diagnosis among cases in the neurological complications dataset was GBS. However, six other neurological conditions were also documented, including encephalitis, myelitis, facial paralysis, meningitis, meningoencephalitis, and optic neuritis. Although some studies have focused exclusively ZIKV-associated GBS [13, 14, 49], others have considered a wider range of neurological conditions linked to recent ZIKV infection [50, 51]. Case reports have described ZIKV-associated myelitis [52], encephalitis [53, 54], meningoencephalitis [55], acute disseminated encephalomyelitis [56], Miller-Fisher syndrome [57], and myasthenia gravis [58]. Some of these reports involved fatalities, young patients, and previously healthy individuals. In addition to patients with ZIKV-associated neurological complications and CZS, studies using human and animal models have accumulated broader evidence that ZIKV is neurotrophic. The virus targets neuronal cell types in the brain, including neural progenitor cells, mature neurons, and astrocytes [59]. ZIKV infection of the central nervous system has been found in both young and adult animals such as mice and non-human primates [59].

Reports of neurological complications associated with ZVD were reported in nearly every department and tended to cluster in large cities with better access to healthcare. This pattern reflects the widespread dissemination of ZIKV throughout Colombia. The city of Barranquilla had the highest number of reported neurological complications. While there was a large ZIKV epidemic in Barranquilla, the city was also subjected to more intensive surveillance for ZIKV-associated neurological complications compared to other cities [60].

There is some agreement between locations with the highest number of ZVD cases and locations with the highest number of ZIKV-associated neurological complications. However, the locations with the highest cumulative incidence of ZIKV per 100,000 population and

locations with the highest incidence of neurological complications per 1,000 ZVD cases are discordant. This mismatch could be due to randomness associated with reporting small numbers of rare events or differences in reporting mild versus severe ZVD cases across the country.

## Conclusions

A strength of this analysis is the quality of the datasets. The ZIKV dataset encompasses the entire duration of the epidemic in Colombia, and all patients in the neurological complications dataset were checked against standardized case definitions. Limitations include lack of detailed clinical information and lack of laboratory confirmation for most ZVD cases. Interestingly, although our report does not include any congenital complications, a recently published report found that out of 5,673 pregnancies with laboratory-confirmed ZVD in Colombia, 2% of infants or fetuses had neurological or eye complications [61]. Another recent study from Colombia found that nine out of 60 children (15%) with laboratory-confirmed ZIKV infection at ages 1–12 months had adverse outcomes on neurologic, hearing, or eye examinations at 20–30 months of age. Six of the remaining 47 children (12.8%) had an alert score in the hearing-language domain [62].

Neurological complications and deaths due to ZIKV were rare in this epidemic. However, more awareness about these risks is needed for people living in or traveling to ZIKV-affected areas. While GBS is relatively easy for non-neurologists to identify, variants such as Miller-Fisher syndrome may not be [57]. Future research should investigate long-term patient outcomes as well as the pathophysiology of these conditions, which can improve treatment strategies [53]. To fully understand the burden of ZIKV, surveillance should encompass a broader spectrum of neurological symptoms of ZVD beyond GBS and microcephaly. Surveillance should also focus on young children, considering the neurotropism of the virus and its effects on postnatal development.

## Acknowledgments

The authors would like to thank all of the medical and public health professionals involved in the reporting of ZVD cases to Sivigila, Colombia's national public health surveillance system.

## Author Contributions

**Conceptualization:** Kelly Charniga, Diana M. Walteros, Marcela Mercado, Pierre Nouvellet, Christl A. Donnelly.

**Data curation:** Zulma M. Cucunubá, Diana M. Walteros, Marcela Mercado, Franklyn Prieto, Martha Ospina.

**Formal analysis:** Kelly Charniga.

**Investigation:** Kelly Charniga.

**Methodology:** Kelly Charniga.

**Software:** Kelly Charniga.

**Supervision:** Pierre Nouvellet, Christl A. Donnelly.

**Visualization:** Kelly Charniga.

**Writing – original draft:** Kelly Charniga.

**Writing – review & editing:** Kelly Charniga, Zulma M. Cucunubá, Diana M. Walteros, Marcela Mercado, Franklyn Prieto, Martha Ospina, Pierre Nouvellet, Christl A. Donnelly.

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
