## [Decision Letter · Decision Letter 0]

2 Mar 2021

PONE-D-21-02674

Descriptive analysis of surveillance data for Zika virus disease and Zika virus-associated neurological complications in Colombia, 2015-2017

PLOS ONE

Dear Dr. Charniga,

Thank you for submitting your manuscript to PLOS ONE. After careful consideration, we feel that it has merit but does not fully meet PLOS ONE’s publication criteria as it currently stands. Therefore, we invite you to submit a revised version of the manuscript that addresses the points raised during the review process.

We look forward to receiving your revised manuscript.

Kind regards,

Berlin Londono

Academic Editor

PLOS ONE

Journal Requirements:

3. We note that Figures 2 and 6 in your submission contain map images which may be copyrighted. All PLOS content is published under the Creative Commons Attribution License (CC BY 4.0), which means that the manuscript, images, and Supporting Information files will be freely available online, and any third party is permitted to access, download, copy, distribute, and use these materials in any way, even commercially, with proper attribution. For these reasons, we cannot publish previously copyrighted maps or satellite images created using proprietary data, such as Google software (Google Maps, Street View, and Earth). For more information, see our copyright guidelines: http://journals.plos.org/plosone/s/licenses-and-copyright.

3.1.    You may seek permission from the original copyright holder of Figures 2 and 6 to publish the content specifically under the CC BY 4.0 license. 

3.2.    If you are unable to obtain permission from the original copyright holder to publish these figures under the CC BY 4.0 license or if the copyright holder’s requirements are incompatible with the CC BY 4.0 license, please either i) remove the figure or ii) supply a replacement figure that complies with the CC BY 4.0 license. Please check copyright information on all replacement figures and update the figure caption with source information. If applicable, please specify in the figure caption text when a figure is similar but not identical to the original image and is therefore for illustrative purposes only.

Reviewers' comments:

Reviewer's Responses to Questions

**Comments to the Author**

1. Is the manuscript technically sound, and do the data support the conclusions?

Reviewer #1: Yes

2. Has the statistical analysis been performed appropriately and rigorously? 

Reviewer #1: Yes

3. Have the authors made all data underlying the findings in their manuscript fully available?

Reviewer #1: Yes

4. Is the manuscript presented in an intelligible fashion and written in standard English?

Reviewer #1: Yes

5. Review Comments to the Author

Reviewer #1: This study sought to describe the prevalence of neurological complications of Zika virus disease (ZVD) in Colombia reported during the 2015-2017 epidemic. The authors used an extensive and large dataset to address question of neurological complications from ZVD. Considering that Zika virus is now endemic in many countries, the current study provides important insights to understanding the risk of ZVD for those living in and traveling to those areas. The authors are to be commended for their thorough study and summary of the data, however, I have a few comments which I would like for the authors to consider before publication. My comments are outlined below.

1. The authors should describe the evidence for Zika virus (ZKV) being neurotrophic. Evidence of ZKV neurotropism is not only found in patients with Guillain-Barré syndrome (GBS) or congenital zika virus syndrome (CZS), but there is preclinical evidence from animal studies showing that zika virus can infect the central nervous system in both young and adult animals (rodents and nonhuman primates). Explaining the evidence for neurotropism can help to explain why more neurologic symptoms beyond GBS should be investigated after ZKV infection in patients (Discussion pg 16 ln 371-379).

2. According to the current study data the highest incidents of ZVD is shown in adults. However, the data also shows that children of both sexes are equally susceptible to ZVD. In fact, a recent prospective study of children in Columbia (Pacheco et al, 2021, Paediatr Perinat Epidemiol, 35: 92-97) demonstrates that 15% of children infected with Zika virus between 1-12 month of age had adverse neurologic, hearing or eye examinations at 20 - 30 months of age. An additional 12.8% received an alert score in the hearing domain. Considering the neurotropism of the virus and the considerable postnatal development from birth to young adulthood, it is important to include broader surveillance of this young population other than severe neurological complications (see also recent review Raper & Chahroudi, 2021, Trop Med Infect Dis, 6 (10): 1-12).

3. Discussion pg 14 Ln 331-334, the authors should briefly mention that the possibility of prior immunity is also not possible due to the timing and origins of the virus arriving in Latin America from French-Polynesia.

4. Discussion pg 14 Ln 344-346, if spending more time at home is an increased risk of mosquito bite and ZVD in older females, would this also be the case for young children?

6. PLOS authors have the option to publish the peer review history of their article (what does this mean?). If published, this will include your full peer review and any attached files.

Reviewer #1: No

---

## [Author Response · Author response to Decision Letter 0]

21 Apr 2021

Regarding data availability, we have revised our statement in lines 432-440: 

The neurological complications dataset cannot be shared publicly as the data contain information on a small number of patients. Although the data are anonymized, identification is a risk given the high geographic resolution and large combination of predictor variables. This determination was made by Comité de Ética y Metodologías de Investigación (CEMIN). To request access to these data, please contact: secretariactin-cein@ins.gov.co, (57+1) 2207700 Ext. 1331-1108, Colombia. The aggregated ZIKV dataset is available on GitHub: https://github.com/kcharniga/descriptive_zika. The data include the number of weekly reported cases by administrative level 2 (municipality) as well as sex and age category.

The GitHub repository is currently set to private mode. If the paper is accepted for publication, the GitHub repository will be made public. 

Regarding Figures 2 and 6, the maps were originally created using GADM version 2. However, we have now replaced them with maps created using shapefiles from the Humanitarian Data Exchange (https://data.humdata.org). These data are available under a Creative Commons Attribution for Intergovernmental Organizations (CC BY-IGO) license (https://data.humdata.org/about/license), which is compatible with CC BY 4.0. The figure legends have been updated accordingly.

In response to Reviewer 1:

Comment 1: The authors should describe the evidence for Zika virus (ZKV) being neurotrophic. Evidence of ZKV neurotropism is not only found in patients with Guillain-Barré syndrome (GBS) or congenital zika virus syndrome (CZS), but there is preclinical evidence from animal studies showing that zika virus can infect the central nervous system in both young and adult animals (rodents and nonhuman primates). Explaining the evidence for neurotropism can help to explain why more neurologic symptoms beyond GBS should be investigated after ZKV infection in patients (Discussion pg 16 ln 371-379).

We fully agree with this comment and have added additional information about the neurotropism of ZIKV to lines 390-395.

Comment 2: According to the current study data the highest incidents of ZVD is shown in adults. However, the data also shows that children of both sexes are equally susceptible to ZVD. In fact, a recent prospective study of children in Columbia (Pacheco et al, 2021, Paediatr Perinat Epidemiol, 35: 92-97) demonstrates that 15% of children infected with Zika virus between 1-12 month of age had adverse neurologic, hearing or eye examinations at 20 - 30 months of age. An additional 12.8% received an alert score in the hearing domain. Considering the neurotropism of the virus and the considerable postnatal development from birth to young adulthood, it is important to include broader surveillance of this young population other than severe neurological complications (see also recent review Raper & Chahroudi, 2021, Trop Med Infect Dis, 6 (10): 1-12).

We would like to thank the reviewer for suggesting these references. In lines 94-97, we have added additional information about postnatal transmission of ZIKV. We have also added the findings from the Pacheco et al. study to lines 417-420. 

Comment 3: Discussion pg 14 Ln 331-334, the authors should briefly mention that the possibility of prior immunity is also not possible due to the timing and origins of the virus arriving in Latin America from French-Polynesia.

We thank the reviewer for this comment and have made the adjustment to lines 338-339.

Comment 4: Discussion pg 14 Ln 344-346, if spending more time at home is an increased risk of mosquito bite and ZVD in older females, would this also be the case for young children?

The reviewer makes a valid point. We agree, and have expanded our discussion of household-based exposure to mosquito bites in lines 356-359.

---

## [Decision Letter · Decision Letter 1]

12 May 2021

Descriptive analysis of surveillance data for Zika virus disease and Zika virus-associated neurological complications in Colombia, 2015-2017

PONE-D-21-02674R1

Dear Dr. Charniga,

We’re pleased to inform you that your manuscript has been judged scientifically suitable for publication and will be formally accepted for publication once it meets all outstanding technical requirements.

Kind regards,

Berlin Londono

Academic Editor

PLOS ONE

Additional Editor Comments (optional):

Reviewers' comments:

Reviewer's Responses to Questions

**Comments to the Author**

1. If the authors have adequately addressed your comments raised in a previous round of review and you feel that this manuscript is now acceptable for publication, you may indicate that here to bypass the “Comments to the Author” section, enter your conflict of interest statement in the “Confidential to Editor” section, and submit your "Accept" recommendation.

Reviewer #1: All comments have been addressed

Reviewer #2: All comments have been addressed

2. Is the manuscript technically sound, and do the data support the conclusions?

Reviewer #1: Yes

Reviewer #2: Yes

3. Has the statistical analysis been performed appropriately and rigorously? 

Reviewer #1: Yes

Reviewer #2: Yes

4. Have the authors made all data underlying the findings in their manuscript fully available?

Reviewer #1: Yes

Reviewer #2: No

5. Is the manuscript presented in an intelligible fashion and written in standard English?

Reviewer #1: Yes

Reviewer #2: Yes

6. Review Comments to the Author

Reviewer #1: I thank the authors for thoroughly addressing all of my comments and congratulations on this excellent work.

Reviewer #2: The manuscript is well written and conclusions are supported by the data. Authors have also addressed all the reviewer's questions satisfactorily.

7. PLOS authors have the option to publish the peer review history of their article (what does this mean?). If published, this will include your full peer review and any attached files.

Reviewer #1: No

Reviewer #2: No

---

## [Editor Report · Acceptance letter]

21 May 2021

PONE-D-21-02674R1 

Descriptive analysis of surveillance data for Zika virus disease and Zika virus-associated neurological complications in Colombia, 2015-2017 

Dear Dr. Charniga:

I'm pleased to inform you that your manuscript has been deemed suitable for publication in PLOS ONE. Congratulations! Your manuscript is now with our production department. 

Kind regards, 

on behalf of

Dr. Berlin Londono 

Academic Editor

PLOS ONE